# Supplementation with a Carob (*Ceratonia siliqua* L.) Fruit Extract Attenuates the Cardiometabolic Alterations Associated with Metabolic Syndrome in Mice

**DOI:** 10.3390/antiox9040339

**Published:** 2020-04-21

**Authors:** María de la Fuente-Fernández, Daniel González-Hedström, Sara Amor, Antonio Tejera-Muñoz, Nuria Fernández, Luis Monge, Paula Almodóvar, Laura Andrés-Delgado, Luis Santamaría, Marin Prodanov, Antonio Manuel Inarejos-García, Angel Luis García-Villalón, Miriam Granado

**Affiliations:** 1Departamento de Fisiología, Facultad de Medicina, Universidad Autónoma de Madrid, 28029 Madrid, Spain; maria.delafuente@uam.es (M.d.l.F.-F.); dgonzalez@pharmactive.eu (D.G.-H.); sara.amor@uam.es (S.A.); antoniotemu@gmail.com (A.T.-M.); nuria.fernandez@uam.es (N.F.); luis.monge@uam.es (L.M.); angeluis.villalon@uam.es (A.L.G.-V.); 2Pharmactive Biotech Products S.L. Parque Científico de Madrid, 28049 Madrid, Spain; palmodovar@pharmactive.eu (P.A.); aminarejos@hotmail.com (A.M.I.-G.); 3Departamento de Química Física Aplicada, Facultad de Ciencias, CIAL (CEI, CSIC-UAM), Universidad Autónoma de Madrid, 28049 Madrid, Spain; marin.prodanov@uam.es; 4Departamento de Anatomía, Histología y Neurociencia, Facultad de Medicina, Universidad Autónoma de Madrid, 28029 Madrid, Spain; laura.andresd@uam.es (L.A.-D.); luis.santamaria@uam.es (L.S.); 5CIBER Fisiopatología de la Obesidad y Nutrición, Instituto de Salud Carlos III, 28006 Madrid, Spain

**Keywords:** metabolic syndrome, carob, cardiovascular, insulin resistance, endothelial dysfunction, coronary ischemia, antioxidant, anti-inflammatory

## Abstract

The incidence of metabolic syndrome (MetS) is increasing worldwide which makes necessary the finding of new strategies to treat and/or prevent it. The aim of this study was to analyze the possible beneficial effects of a carob fruit extract (CSAT+^®^) on the cardiometabolic alterations associated with MetS in mice. 16-week-old C57BL/6J male mice were fed for 26 weeks either with a standard diet (chow) or with a diet rich in fats and sugars (HFHS), supplemented or not with 4.8% of CSAT+^®^. CSAT+^®^ supplementation reduced blood glucose, Homeostatic Model Assessment of Insulin Resistance (HOMA-IR) and circulating levels of total cholesterol, low-density lipoprotein (LDL) cholesterol (LDL-c), insulin, and interleukin-6 (IL-6). In adipose tissue and skeletal muscle, CSAT+^®^ prevented MetS-induced insulin resistance, reduced macrophage infiltration and the expression of pro-inflammatory markers, and up-regulated the mRNA levels of antioxidant markers. Supplementation with CSAT+^®^ prevented MetS-induced hypertension and decreased the vascular response of aortic rings to angiotensin II (AngII). Moreover, treatment with CSAT+^®^ attenuated endothelial dysfunction and increased vascular sensitivity to insulin. In the heart, CSAT+^®^ supplementation reduced cardiomyocyte apoptosis and prevented ischemia-reperfusion-induced decrease in cardiac contractility. The beneficial effects at the cardiovascular level were associated with a lower expression of pro-inflammatory and pro-oxidant markers in aortic and cardiac tissues.

## 1. Introduction

Metabolic syndrome (MetS) is a pathologic condition characterized by abdominal obesity, insulin resistance, hypertension, and hyperlipidemia. Among the risk factors to suffer MetS are age, genetic predisposition, and unhealthy lifestyle habits, which include the consumption of a diet rich in carbohydrates and saturated fats and a sedentary lifestyle [1,2].

Main associated pathologies with MetS are type-2 diabetes and cardiovascular diseases (CVD), being reported that suffering MetS increases the risk of developing both CVD and diabetes by two and five times, respectively [3,4].

Although there is no consensus about the etiology of MetS, it is clear that the state of chronic low-grade inflammation plays a major role in both the initiation of the syndrome and in the development of its associated pathologies [5,6]. The inflammatory state may have its origin in visceral adipose tissue as a result of both structural and functional alterations of adipocytes which include decreased plasticity, hypertropia, hyperplasia, and inability to buffer the excess nutrient intake derived from the positive energetic balance, which results in a dyslipidemic state and in the development of insulin resistance [5]. The chronic exposure to inflammatory cytokines and fatty acids induces alterations in the activation of intracellular pathways in several organs and tissues. Particularly, in insulin-dependent tissues such as skeletal muscle and adipose tissue, there is a reduction in the activation of the phosphoinositide 3-kinase (PI3K)/Akt pathway which derives in decreased translocation of glucose transporter 4 (GLUT-4) to the cell membrane and the consequent reduction in glucose uptake and hyperglycemia [6].

In addition to type-2 diabetes, MetS is also associated with cardiovascular alterations both at the cardiac and at the vascular level [7]. These alterations include activation of the renin-angiotensin-aldosterone system [8] and activation of the sympathetic nervous system [9], which result in increased renal sodium retention, plasmatic volume, blood pressure, heart rate, and cardiac output [10]. In the heart, left ventricular hypertrophy is reported [11]. Importantly, these alterations have been associated with reduced diastolic and systolic contractile function, both in humans and in experimental animals with MetS and take place prior to any evidence of atherosclerotic disease [7]. Moreover, cardiac responses to certain pathologic conditions, such as myocardial ischemia, are also known to be significantly affected by MetS [12,13]. Particularly, MetS impairs the balance between coronary blood flow and myocardial metabolism in response to cardiac ischemia [14].

The cardiovascular alterations associated with MetS may be related, at least in part, to alterations in cardiovascular insulin sensitivity as insulin exerts important effects in the cardiovascular system. In physiological conditions, insulin promotes nitric oxide-mediated arterial vasodilation through the activation of the intracellular PI3K/Akt pathway and endothelin-1 mediated arterial vasoconstriction through the activation of the mitogen-activated protein kinase (MAPK) in the vascular endothelium [15]. In states of insulin resistance, there is a reduction in the activation of the PI3K/Akt pathway whereas the activation of the MAPK pathway remains either unaffected or hyper activated, leading to an imbalance in the vasodilator and vasoconstrictor effects of insulin that may contribute to the development of hypertension and other cardiovascular complications [16].

In the last years the incidence of MetS and its associated co-morbidities have extremely increased, not only in developed countries where it affects approximately 20% of the population [1], but also in developing countries [2]. Thus, it is a priority to find therapeutic approaches for its treatment and prevention. The lack of efficacy of pharmacological therapies added to the side effects [17], and has led to the search for new therapeutic alternatives of natural origin with less adverse effects.

Among them the fruit of carob tree, (*Ceratonia siliqua* L.), may be a good candidate since several studies have reported that it exerts beneficial effects improving the glycemic status [18,19,20,21], the lipid profile [22,23,24,25,26], and attenuating the cardiovascular alterations associated with MetS [23,27,28]. Its therapeutic effects are attributed to several bioactive compounds that are present not only in the pulp but also in the pod and the seeds of the carob fruit. These compounds are sugars like sucrose, glucose, and fructose (48–56%), dietary fibers like cellulose, hemicellulose, and lignin (30–40% in carob’s pulp), gum (85% of carob seed is composed of galactomannan), amino acids, minerals (mainly potassium and calcium), and phenolic compounds [29].

Although carob pulp is reported to exert several biological effects mainly because of the presence of phenolic compounds [26,30]. In this study we have focused in analyzing the effects of supplementation with a carob pod and seeds extract, since it is reported that these parts contain higher amounts of galactomanns, gallic acid, and gallotannins and exert a stronger antioxidant capacity as well as anti-hypertensive effects in vitro [31]. Thus, the aim of this study is to analyze the possible beneficial effects of supplementation with a carob pod and seed extract (CSAT+^®^) in the cardiometabolic alterations associated to MetS in mice. Particularly we analyzed the in vivo effects of supplementation with CSAT+^®^ on metabolism, including insulin sensitivity and lipid profile, and cardiovascular function in mice fed a high fat/high sucrose diet (HFHS).

## 2. Material and Methods

### 2.1. Materials

Samples of proprietary blend of carob (*Ceratonia silique* L.) pods and seeds extracts together with fructooligosaccharides (FOS) marketed under the brand CSAT+^®^ were provided by Pharmactive Biotech Products S.L (Madrid, Spain). All of them were in a fluid powder form and were stored in darkness until addition into the animal chow. All carob samples were standardized to 36–40% of galactomannan fiber, 2–4% of FOS and ≥ 1% total polyphenols by UV.

### 2.2. Chemical Characterization of CSAT+^®^ Samples

#### 2.2.1. Analysis of Phenolic Compounds by RP-HPLC-PAD/MS

Identification of the phenolic compounds were carried out by reversed-phase high performance liquid chromatography (RP-HPLC) coupled to an electrospray ionization mass spectrometry (ESI-MS) detector, according to the method proposed by Kammerer et al. 2004 [32]. Briefly, the chromatographic system was an Agilent 1200 series, (Santa Clara, CA, USA). Consists of a binary pump, autosampler and a photodiode array detector (PAD)) coupled to an ESI source and quadrupole mass analyzer. The mobile phase was pumped at a flow rate of 1 mL/min. It was a linear gradient of eluent A (2% (*v*/*v*) acetic acid in water), and eluent B (0.5% acetic acid in water and acetonitrile 50:50 (*v*/*v*)) as follows: from 0 to 20 min, 10% to 24% of B; from 20 to 40, 24% to 30 of B; from 40 to 60 min, 30% to 55% of B; from 60 to 75 min, 55% to 100% of B; during 8 min, maintaining 100% of B; from 83 to 85 min, from 100% to 10% of B and during 5 min maintaining 10% of B. The total run time was 90 min. The PAD was set at 280 and 370 nm and the injection volume was 20 µL. ESI-MS was tuned as follows: mass was recorded in the range *m*/*z* < 200 at 100 V, in the range *m*/*z* 200–1000 at 200 V and in the range *m*/*z* 1000–2500 at 250 V, negative ionization mode, a drying gas flow of 10.0 L/min of N_2_ at 340 °C, with a nebulizer pressure of 40 psi, and capillary tension of 4000 V.

Chromatographic peaks were identified tentatively on the base of their order of elution, ultraviolet-visible spectra, pseudo-molecular ions, and diagnostic fragments already described in the literature [32,33,34,35,36].

#### 2.2.2. Total Antioxidant Capacity by ABTS Decolorization Assay

For assessment of total antioxidant capacity the method based on 2,2′-azino-bis(3-ethylbenzothiazoline-6-sulfonic acid (ABTS) decolorization was performed according to Oki et al. 2006 [35]. An ABTS^+^ aqueous stock solution was prepared by adding 140 mM potassium persulfate (44 µL) to 2.5 mL of 7 mM ABTS^−^ (#A1888; Sigma-Aldrich, St. Louis, MO, USA), and the mixture was then allowed to stand for 16 h at room temperature. The working solution of the radical ABTS^+^ was prepared by diluting the stock solution 1:75 (*v*/*v*) in 5 mM sodium phosphate buffer (pH 7.4) to obtain an absorbance value of 0.7 ± 0.02 at 734 nm. Samples (30 µL) were added to 270 µL working ABTS^+^ solution in a microplate. The absorbance was assessed at 734 nm for 10 min at 30 °C with measurements at every 2 min. After 5 min, the reaction was complete. Trolox (#238813; Sigma-Aldrich, St. Louis, MO, USA) calibration curve was used for quantification. Results were expressed as % of Trolox equivalents (dry weight). All measurements were performed in triplicate.

### 2.3. In Vivo Study

#### 2.3.1. Animals

All the experiments were conducted according to the European Union Legislation and with the approval of the Animal Care and Use Committee of the Community of Madrid (PROEX 039/18).

Thirty-six 16-week-old C57/BL6J mice were housed two per cage and maintained in climate-controlled quarters with a 12 h light cycle and under controlled conditions of humidity (50–60%) and temperature (22–24 °C). Mice were fed ad libitum and divided in three experimental groups: mice fed with a standard chow (Chow; *n* = 12); mice fed a high fat/high sucrose (HFHS) diet (Research Diets 12331-1) containing 58% kcal from fat with sucrose (HFHS; *n* = 12); and mice fed a high fat/high sucrose (HFHS) diet (Research Diets 12331-1) containing 58% kcal from fat with sucrose supplemented with 4.8% of CSAT+^®^ (HFHS + CSAT+^®^; *n* = 12). All mice were maintained on diets for 26 weeks. A weekly control of body weight and solid and liquid intake was performed. At the age of 9 months, all animals were injected an overdose of sodium pentobarbital (100 mg/kg) and killed by decapitation after overnight fasting.

After euthanasia, blood was collected from the decapitated trunk in tubes with EDTA (1.5 mg/mL) and centrifuged at 3000 rpm for 20 min to obtain the plasma. Retroperitoneal visceral, lumbar subcutaneous, and brown interescapular adipose tissue depots as well as hypothalamus, kidneys, adrenal glands, spleen, liver, gastrocnemius, and soleus muscles were immediately removed and weighed. All tissues were stored at −80 °C for later analysis.

In addition, a slice of gastrocnemius muscle, retroperitoneal adipose tissue, and heart were fixed overnight in 4% paraformaldehyde and embedded in paraffin for further histological analysis.

#### 2.3.2. Glucose Tolerance Test (GTT) and Homeostatic Model Assessment of Insulin Resistance (HOMA-IR)

One week before euthanasia, all animals were subjected to a glucose tolerance test (GTT). After overnight fasting, mice were administered an intraperitoneal (i.p.) bolus of glucose (2 mg/kg. Glycemia was measured by venous tail puncture using Glucocard^TM^ G (Arkray Factory, Inc., Koji Konan-cho, Koka, Shiga, Japan) 5 min before (basal glycaemia) and 30, 60, 120, and 150 min after glucose injection. The total area under the curve (AUC) for the glucose response was calculated with the following formula: AUC = 25 × (fasting value) + 0.5 × (30 min value) + 0.75 × (1 h value) + 0.5 × (2 h value) [37]. The Homeostatic Model Assessment of Insulin Resistance (HOMA-IR) index was calculated through the following formula: fasting glucose (mg/dL) × (fasting insulin (ng/m)/405) as previously described [38].

#### 2.3.3. Serum Measurement

##### Metabolic Hormones

Serum concentrations of leptin, insulin, and adiponectin were measured by ELISA kits (Merck Millipore, Dramstadt, Germany) following the manufacturer’s instructions. The sensitivity of the method for leptin, insulin, and adiponectin was 0.04, 0.2, and 0.16 ng/mL, respectively. The intraassay variation was between 1.9–2.5% for leptin, 0.9–8.4% for insulin, and 0.43–1.96% for adiponectin.

##### Lipid Profile

Triglycerides, total cholesterol, low-density lipoprotein (LDL), and high-density lipoprotein (HDL) were measured in the serum using commercial kits from Spin React S.A.U (Sant Esteve de Bas, Gerona, Spain) following the manufacturer’s instructions.

##### Pro-Inflammatory Cytokines

Interleukin-6 (IL-6) plasma levels were measured by an ELISA kit (Cusabio, Wuhan, China) following the manufacturer’s instructions. The sensitivity of the method was 0.078 pg/mL and the intraassay and interassay variations were <8% and <10%, respectively.

#### 2.3.4. Measurement of Mean Arterial Pressure in Conscious Mice by the Tail-Cuff System

Mean arterial blood pressure (MBP) measurements were performed in each mouse every other day for two weeks before sacrifice by tail-cuff plethysmography using a Niprem 645 blood pressure system (Cibertec, Madrid, Spain). For that purpose, mice were placed in a quiet area (22 ± 2 °C) and habituated to the experimental conditions for at least 3 days. Before measurements, mice were prewarmed to 34 °C for 10–15 min. Then, the occlusion cuff was placed at the base of the tail and the sensor cuff was placed next to the occlusion cuff. Next, the occlusion cuff was inflated to 250 mm Hg and deflated over 20 s. Five to six measurements were recorded in each mouse and the mean of all measurements was calculated each day per animal.

#### 2.3.5. Reactive Hyperemia

After 12 weeks of treatment mice were anesthetized with a continuous flow of 1.5 isoflurane in 100% oxygen and maintained in decubitus position with limbs fixed in extension with transparent tape. Cutaneous blood perfusion of the footpad was recorded using Laser Speckle Contrast Imaging (Moor FLPI2 system, Moor Instruments, Axminster, UK) with a laser wave length of 785 nm. The laser head was positioned 20 cm above the skin. The acquisition rate was 2 frames per second. After 20 min of flow stabilization, footpad blood perfusion was studied during and after 3 min occlusion through a tourniquet at the groin. The repayment/debt ratio was calculated as the ratio of the AUC (area under curve) of the post-occlusive hyperemia and the AUC of perfusion debt during ischemia due to vascular occlusion.

#### 2.3.6. Experiments of Vascular Reactivity

For the vascular reactivity experiments, the aorta was carefully dissected, cut in 2 mm segments and kept in cold isotonic saline solution. Thoracic segments were used for the vasodilation studies and abdominal segments for the vasoconstriction studies. The assembly of the segments was performed as previously described [39,40]. Changes in isometric force were recorded using a PowerLab data acquisition system (ADInstruments, Colorado Springs, CO, USA). After applying an optimal passive tension of 1 g, vascular segments were allowed to equilibrate for 60–90 min. Afterwards, segments were stimulated with potassium chloride solution (100 mM KCl, 7447-40-7 Merck Millipore, Burlington, MA, USA) to determine the contractility of smooth muscle. Segments which failed to contract at least 0.5 g of KCl were discarded. For the vasodilation experiments, the segments were precontracted with U46619 10^−7.5^ M (#D8174, Sigma-Aldrich, St. Louis, MO, USA) prior to adding a single dose of insulin (10^−6^ M) or to perform dose-response curves in response to acetylcholine (10^−9^–10^−4^ M) and sodium nitroprusside (NTP; 10^−9^–10^−5^ M) (#I0516, #A6625 and #71778, Sigma-Aldrich, St. Louis, MO, USA).

To study the effect of oxidative stress in the endothelium, some segments were preincubated for 30 min with tempol 10^−4^ M and catalase 10.000 U/mL ((#CAS 2226-96-2 and #C-9322, Sigma-Aldrich, St. Louis, MO, USA) before the dose-response curves in response to acetylcholine.

In order to study the mechanism of insulin response, some segments were pre-incubated for 30 min with the inhibitor of the nitric oxide synthaseL-NAME (10^−4^ M) (Nω-nitro-L-arginine methyl ester hydrochloride) (#N5751, Sigma-Aldrich, St. Louis, MO, USA), the inhibitor of cyclooxygenase indomethacin (10^−5^ M) (#53-86-1, Sigma-Aldrich, St. Louis, MO, USA) or the inhibitor of endothelin-1 receptor type A BQ-123 (10^−6^ M) (#136553-81-6, Sigma-Aldrich, St. Louis, MO, USA). For each dose–response curve, the logarithm of the concentration producing 50% of the maximal response (ED50) was calculated by geometric interpolation.

For the vasoconstriction studies dose-response curves were performed in response to norepinephrine (10^−9^–10^−4^ M), endothelin-1 (10^−10^–10^−6.5^M), and angiotensin-II (10^−11^–10^−6^M) (#A7257, #E7764 and #A9525, Sigma-Aldrich, St. Louis, MO, USA). Contraction in response to vasoconstrictors was represented as % contraction to KCl.

#### 2.3.7. Experiments of Heart Perfusion: Langendorff

After euthanasia, hearts were immediately removed and mounted in the Langendorff perfusion system as previously described [40]. Briefly, after a 30 min equilibration period with constant flow perfusion, global ischemia was induced by stopping the flow perfusion for 30 min. Afterwards, hearts were re-perfused for 45 min. Coronary perfusion pressure was measured through a lateral connection in the perfusion cannula and left ventricular pressure was measured using a latex balloon inflated to a diastolic pressure of 5–10 mmHg, both of them connected to Statham transducers (Statham Instruments, Los Angeles, CA, USA). Left ventricular pressure was recorded and was used to calculate the first derivative of the left ventricular pressure curve (dP/dt) as an index of heart contractility and the heart rate. These parameters were recorded on a computer using the PowerLab/8e data acquisition system (ADInstruments, Colorado Springs, CO, USA).

After ischemia-reperfusion (IR), hearts were collected and stored at −80 °C for further analysis.

#### 2.3.8. RNA Extraction and Quantitative Real-Time Polymerase Chain Reaction (qPCR)

Total RNA was extracted from 100 mg of retroperitoneal adipose tissue, gastrocnemius muscle, heart, and aortic tissue using the Tri-Reagent protocol [41]. cDNA was then synthesized from 1 µg of total RNA using a high-capacity cDNA RT kit (Applied Biosystems, Foster City, CA, USA). mRNA levels were assessed by using assay on-demand kits (Applied Biosystems) for each gene. TaqMan Universal PCR Master Mix (Applied Biosystems, Foster City, CA, USA) was used for amplification according to the manufacturer’s protocol in a Step One System (Applied Biosystems, Foster City, CA, USA).

In retroperitoneal adipose tissue and gastrocnemius muscle the gene expression of interleukin-6 (IL-6) (Mm00446190_m1), interleukin-1 beta (IL-1β) (Mm00434228_m1), tumor necrosis factor-alpha (TNF-α) (Mm00443258_m1), monocyte chemoattractant protein (MCP-1) (Mm00441242_m1), glutathione reductase (GSR) (Mm00439154_m1), NADPH oxidase-4 (NOX-4) (Mm00479246_m1), and lipoxygenase (LOX) (Mm00507789_m1) were analyzed. IL-1β, IL-6, cyclooxygenase 2 (COX-2) (Mm03294838_g1), MCP-1, transforming growth factor beta (TGFβ) (Mm01227699_m1), superoxide dismutase 1 (SOD-1) (Mm01344233_g1), LOX, GSR, endothelial nitric oxide synthase (eNOS) (Mm00435217_m1), and inducible nitric oxide synthase (iNOS) (Mm00440502_m1) were measured in heart samples. Finally, in aortic tissue, the mRNA concentrations of IL-1β, IL-6, COX-2, MCP-1, TNFα, TFGβ, iNOS, SOD-1, NOX-4, LOX, GSR, alpha-1 adrenergic receptor (α_1_ adre.) (Mm00442668_m1), endothelin receptor type A (ETA) (Mm01243722_m1), endothelin receptor type B (ETB) (Mm00432989_m1), endothelin-1 (ET1) (Mm00438656_m1), angiotensin receptor type 1 (AT1) (Mm00616371_m1), and angiotensin receptor type 2 (AT2) (Mm00431727_g1) were measured. Values were normalized to the housekeeping 18S (Mm03928990_g1). To determine the relative expression levels the ΔΔC_T_ method was used [42].

#### 2.3.9. Incubation of Aorta Segments, Retroperitoneal Adipose Tissue and Gastrocnemius Muscle Explants in Presence/Absence of Insulin (10^−6^ M)

Thoracic aorta segments and explants of retroperitoneal adipose tissue and gastrocnemius muscle (100 mg) were incubated with Dulbecco’s modified Eagle’s medium Gibco ((DMEM/F-12; 1:1 mix; Invitrogen, Carlsbad, CA, USA) plus 100 U/mL penicillin and 100 μg/mL streptomycin (Invitrogen, Carlsbad, CA, USA) in the presence/absence of insulin (10^−6^ M) (Sigma-Aldrich, St. Louis, MO, USA) at 37 °C in a 95% O_2_ and 5% CO_2_ incubator. After 30 min (aorta) or 15 min (adipose and muscle tissue) of incubation, both the tissue and the culture media were collected and stored at −80 °C for further analysis.

#### 2.3.10. Protein Quantification by Western Blot

Total of 100 mg of retroperitoneal adipose tissue, gastrocnemius or aortic tissue were homogenized using RIPA buffer. After centrifugation (12.000 rpm, 4 °C, 20 min), the supernatant was collected and total protein content was measured by the Bradford method (Sigma-Aldrich, St. Louis, MO, USA). In each assay, 100 μg of total protein was loaded in each well. After electrophoresis using resolving acrylamide SDS gels (10%) (Bio-Rad, Hércules, CA, USA) proteins were transferred to polyvinylidene difluoride (PVDF) membranes (Bio-Rad, Hércules, CA, USA). Transfer efficiency was determined by Ponceau red dyeing (Sigma-Aldrich, St. Louis, MO, USA). Membranes were then blocked either with tris-buffered saline (TBS) containing 5% (*w/v*) non-fat dried milk for non-phosphorylated protein or with 5% BSA for phosphorylated protein and incubated with the appropriate primary antibody for Akt (1:1000; # 04-796, Merk Millipore, Dramstad, Germany) or p-Akt (Ser 473) (1:500; #9271, Cell Signaling Technology, Danvers, MA, USA). Membranes were subsequently washed and incubated with the secondary antibody conjugated with peroxidase (1:2000; Pierce, Rockford, IL, USA). Peroxidase activity was visualized by chemiluminescence and quantified by densitometry using BioRad Molecular Imager ChemiDoc XRS System (Hércules, CA, USA).

Afterwards, membranes were also incubated with a primary antibody against GAPDH (1:1000; Ambion Life Technologies, Waltham, MA, USA) and its respective secondary antibody in order to normalize each sample for gel-loading variability. For each sample relative protein expression levels were calculated in relation to protein expression levels in chow mice.

#### 2.3.11. Immunohistochemistry

Gastrocnemius muscle and adipose samples were fixed in 4% paraformaldehyde (PFA) in phosphate-buffered saline (PBS) overnight. Samples were then washed in PBS, dehydrated and paraffin wax embedded. Sections (5 μm) were cut on a Microtome, mounted on Superfrost slides and dried overnight at 37 °C. Sections were deparaffinized in xylol, rehydrated and washed in distilled water. Endogenous peroxidase was blocked with 3% H_2_O_2_ and incubated 3 min with Proteinase K (Roche, Basilea, Switzerland) for antigen retrieval. Several washing steps were followed by blocking for 1 h with 5% BSA in PBS followed by overnight incubation at 4 °C with a primary antibody that detects macrophages: F4/80 (#MCA497GA, Bio-Rad, Hercules, California, USA) at 1:100. Biotin- conjugated secondary antibody (Thermo Fisher Scientific, Hampton, NH, USA) was diluted in PBS at 1:1000 and incubated for 25 min. Streptavidin (Thermo Fisher Scientific, Hampton, NH, USA) was added and used to revealed primary antibody signal with diaminobenzidine (DAB). Nuclei were counterstained with hematoxylin for 1 min.

The macrophage density was estimated by dividing the total number of positive cells by the total volume analyzed using an optical dissector in a stereological light microscope (Olympus, Tokyo, Japan) and NewCAST stereological software package (Visiopharm, Horsholm, Denmark). Images were acquired using a microscope fitted with a 63 × 1.4 NA objective.

#### 2.3.12. TUNEL Assay

DNA fragmentation, a marker of apoptotic cell death, was evaluated using the TUNEL staining according to the manufacturer’s instructions (Sigma-Aldrich, St. Louis, MO, USA). Hearts were fixed in 4% paraformaldehyde (PFA) in phosphate-buffered saline (PBS) overnight. Samples were then washed in PBS, dehydrated and paraffin wax embedded. Sections (5 μm) were cut on a Microtome, mounted on Superfrost slides and dried overnight at 37 °C. Sections were deparaffinized, rehydrated, and washed in distilled water. Epitopes were retrieved by heating in citrate buffer (pH 6.0) for 15 min in a microwave at full power. Heart sections were permeabilized with 0.5% Triton X-100 (Panreac, Barcelona, Spain) in PBS for 15 min. Non-specific binding sites were saturated by incubation for at least 1 h in blocking solution (5% BSA, 5% goat serum, 0.3% Tween-20 in PBS). Endogenous biotin was blocked with the avidin-biotin blocking kit (Vector, Burlingame, CA, USA). Several washing steps were followed by incubation with Terminal Deoxynucleotidyl Transferase (TdT) buffer (#16314015 Sigma-Aldrich, St. Luis, MO, USA) for 20 min and then overnight incubation with TdT antibody (#3333566001, Sigma-Aldrich, St. Louis, MO, USA) while protected from exposure to light. Streptavidin-Cy3 (Jackson ImmunoResearch, Cambridgeshire, UK) was used to reveal TUNEL positive cells. Nuclei were stained with DAPI for 30 min and slides were mounted in ProLong Diamond (Thermo Fisher Scientific, Hampton, NA, USA).

Images were acquired using a Leica SP5 confocal microscope fitted with a 63 × 1.4 NA objective (Wetzlar, Germany). Z-stacks were taken every 5 µm. Maximal projections of images were 3D reconstructed in whole-mount views using FIJI for Windows 36bit (NIH, Bethesda, MA, USA). Percentage of TUNEL-positive cells were referred to the total myocardial cell number in samples from each experimental group.

#### 2.3.13. Statistical Analysis

Data are represented as mean ± SEM and were analyzed by one-way or two-way ANOVA (dose-response curves) followed by Bonferroni post-hoc test using GraphPad Prism 5.0 (GraphPad Software, La Jolla, CA, USA). Differences were considered significant when *p* < 0.05.

## 3. Results

### 3.1. Chemical Characterization of CSAT+^®^ by RP-HPLC-PAD-MS

Several phenolic compounds of CSAT+^®^ were identified by HPLC-MS. These include gallic acid, 28 gallotannins, 6 flavonols, 3 flavanones, and 1 ellagic acid conjugate (Table 1).

### 3.2. Total Antioxidant Capacity of CSAT+^®^

CSAT+^®^ exhibited antioxidant capacity regarding ABTS^·^ decolorization assay, showing a 5.5 ± 0.9% as Trolox equivalents.

### 3.3. Body Weight, Daily Food Intake, and Organ Weights

Table 2 shows the body weight, the daily food intake, and the organ weights of visceral retroperitoneal adipose tissue, subcutaneous lumbar adipose tissue, interescapular brown adipose tissue, heart, kidneys, adrenal glands, spleen, liver, gastrocnemius, and soleus muscles of mice from the three experimental groups.

Before starting with the diets there were no differences in body weight between mice (data not shown). After 26 weeks, mice fed with the HFHS diet showed increased body weight and decreased daily food intake compared to chow animals, regardless of whether or not they had been supplemented with CSAT+^®^ (*p* < 0.001 for both) (Table 2). HFHS was associated with a significant increase in the weight of visceral retroperitoneal adipose tissue (*p* < 0.001), subcutaneous lumbar adipose tissue (*p* < 0.001), interscapular brown adipose tissue (*p* < 0.001), heart (*p* < 0.001), kidneys (*p* < 0.01), adrenal glands (*p* < 0.05), spleen (*p* < 0.01), liver (*p* < 0.05), and soleus muscle (*p* < 0.01) (Table 2). CSAT+^®^ supplementation prevented obesity-induced increase of heart and soleus muscle weights (*p* < 0.05 for both) and further increased the amount of interscapular brown adipose tissue (*p* < 0.001) (Table 2).

### 3.4. Glycaemic Status, Lipid Profile, and Plasma Concentrations of Metabolic Hormones

Figure 1 shows the circulating levels of glucose (A), insulin (B), and the HOMA-IR (C), as well as the glucose levels over time (D) and the AUC (area under the curve) (E) after an i.p. injection of a glucose bolus.

Mice with MetS showed a significant increase of glycaemia, insulin circulating levels, HOMA- IR, and AUC (*p* < 0.001 for all) and these alterations were attenuated by CSAT+^®^ supplementation (*p* < 0.05). Consumption of HFHS diet for 26 weeks also induced a significant increase in the circulating levels of leptin, triglycerides, total cholesterol, total lipids, LDL-c, HDL-c (Table 3; *p* < 0.001 for all), and IL-6 (Table 3; *p* < 0.05), whereas it significantly decreased the plasma concentrations of adiponectin (Table 3; *p* < 0.01). Treatment with CSAT+^®^ increased adiponectin plasma levels (Table 3; *p* < 0.01) and attenuated the obesity-induced increase in total cholesterol and LDL-c (Table 3; *p* < 0.05 for both).

### 3.5. Gene Expression of Inflammatory Markers in Visceral Adipose Tissue and Skeletal Muscle

In retroperitoneal adipose tissue, obese mice showed an overexpression in the mRNA levels of IL-1β (Figure 2A; *p* < 0.001), TNF-α (Figure 2C; *p* < 0.01), and MCP-1 (Figure 2D; *p* < 0.01) whereas the gene expression of IL-6 was unchanged among experimental groups (Figure 2B). The staining with the antibody F4/80 also revealed an increased macrophage infiltration in HFHS fed mice compared to chow animals (Figure 2J; *p* < 0.05). In skeletal muscle only the mRNA levels of MCP-1 (Figure 3D) and macrophage infiltration (Figure 3I,J) were significantly up regulated in response to the HFHS diet (*p* < 0.001 and *p* < 0.05, respectively).

Macrophage infiltration and MCP-1 mRNA levels were significantly reduced after supplementation with CSAT+^®^ for 26 weeks, both in adipose tissue and in skeletal muscle (*p* < 0.05). In addition, treatment with CSAT+^®^ attenuated MetS-induced overexpression of IL-1β (*p* < 0.01) and TNF-α (*p* < 0.05) in adipose tissue (Figure 2I,J).

### 3.6. Gene Expression of Oxidative Stress-Related Markers in Visceral Adipose Tissue and Skeletal Muscle

The gene expression of oxidative stress-related markers in retroperitoneal adipose tissue and gastrocnemius skeletal muscle is shown in Figure 2 and Figure 3 respectively.

In retroperitoneal adipose tissue, no significant changes were found among experimental groups. In skeletal muscle, we found an overexpression in the mRNA levels of NOX-4 (Figure 3G *p* < 0.05) and a downregulation of LO (Figure 3H *p* < 0.05) in response to HFHS diet that was not prevented by CSAT+^®^ treatment. However, CSAT+^®^ supplementation significantly increased the gene expression of the antioxidant enzymes SOD-1 (Figure 3A) and GSR (Figure 3B) in comparison to both chow animals (*p* < 0.001 for both) and untreated HFHS mice (*p* < 0.05 for both).

### 3.7. Activation of the PI3K/Akt Pathway in Visceral Adipose Tissue and Skeletal Muscle Explants in Response to Insulin

In order to assess insulin sensitivity in visceral adipose tissue and skeletal muscle explants of both tissues were incubated in the presence/absence of insulin. In the absence of insulin, we did not find differences in the activation of the PI3K/Akt pathway either in adipose tissue (Figure 2I) or in skeletal muscle explants (Figure 3I) among experimental groups. However, after incubation of retroperitoneal adipose tissue and gastrocnemius explants with insulin for 15 min a significant increase in the ratio p-Akt/Akt both in chow (*p* < 0.05) and in HFHS animals supplemented with CSAT+^®^ (*p* < 0.05) was found, but not in the untreated HFHS mice.

### 3.8. Blood Pressure and Vascular Reactivity in Response to the Vasoconstrictors Noradrenaline (NA), Endothelin-1 (ET-1), and Angiotensin II (AngII)

Compared to chow animals, HFHS diet induced a significant increase in mean arterial pressure (*p* < 0.01) that was totally prevented by CSAT+^®^ treatment (*p* < 0.01) (Figure 4A).

Results from vascular reactivity experiments showed no significant changes in the vascular response of aorta segments to endothelin-1 (ET-1) among experimental groups (Figure 4C). However, a decreased response of aorta segments of HFHS-fed animals to noradrenaline (NA) compared to chow-fed animals was found, regardless of whether or not they had been supplemented with CSAT+^®^ (*p* < 0.05) (Figure 4B). Finally, the vascular response of aorta segments from HFHS + CSAT+^®^ mice to Angiotensin II (Ang II) was significantly reduced compared to aorta segments from both chow and untreated HFHS mice (*p* < 0.05 for both) (Figure 4D).

### 3.9. Assessment of Vascular Function In Vivo by Reactive Hyperemia and Ex Vivo by Endothelium-Dependent and Endothelium-Independent Relaxation of Aorta Segments

Results from reactive hyperemia experiments after 12 weeks of diet and dose response-curves of aorta segments in response to both acetylcholine (Ach) and sodium nitroprusside (NPT) after 26 weeks of diet are shown in Figure 5.

After 12 weeks, untreated HFHS-fed mice showed decreased hyperemic blood flow after occlusion of the femoral artery compared to chow animals (*p* < 0.001) with this effect being significantly attenuated by CSAT+^®^ supplementation (Figure 5A; *p* < 0.05).

After euthanasia, no significant differences were found among experimental groups in the endothelium-independent relaxation of aorta segments in response to NTP (Figure 5B). On the contrary, endothelium-dependent relaxation in response to Ach was significantly decreased in aorta segments from untreated mice with MetS (Figure 5C; *p* < 0.001), an effect that was prevented by CSAT+^®^ supplementation (*p* < 0.05), and by incubation of aorta segments with the antioxidants Tempol and Catalase (Figure 5D; *p* < 0.05). On the contrary, preincubation with antioxidants did not affect Ach-induced relaxation of aorta segments from chow and HFHS-fed mice.

### 3.10. Vascular Reactivity and Activation of PI3K/Akt Pathway in Aorta Segments in Response to Insulin

Activation of PI3K/Akt pathway in aortic tissue in response to insulin, as well as vascular relaxation of aorta segments in response to accumulative doses of insulin in the presence/absence of the inhibitor of the nitric oxide synthase L-NAME (10^−4^ M), the inhibitor of cyclooxygenase, indomethacin (10^−5^ M), or the inhibitor of endothelin receptor type A, BQ-123 (10^−6^ M) are shown in Figure 6.

Incubation of aorta segments of mice from the three experimental groups with culture medium did not affect the activation of the PI3K/Akt pathway (Figure 6A). However, incubation with insulin (10^−6^ M) for 30 min significantly up-regulated the p-Akt/Akt ratio in chow (*p* < 0.05) and HFHS + CSAT+^®^ mice (*p* < 0.05), but not in untreated HFHS animals.

Results from vascular reactivity experiments showed a vasodilator effect of insulin in aorta segments from mice from the three experimental groups (Figure 6B). However, the vasodilator effect in response to insulin was significantly lower in aorta segments from untreated HFHS mice compared to controls (*p* < 0.01), an effect that was prevented by CSAT+^®^ supplementation (*p* < 0.05).

Pre-incubation of aorta segments with L-NAME reduced insulin-induced relaxation of aorta segments in all experimental groups, although this effect was only statistically significant in aorta segments from chow animals (Figure 6C; *p* < 0.05). Pre-incubation with indomethacin did not affect insulin-induced relaxation in aorta segments from chow and untreated HFHS animals, but it significantly increased the insulin-induced vasodilation of aorta segments from HFHS + CSAT+^®^ mice (Figure 6E; *p* < 0.05). Finally, pre-incubation of aorta segments with BQ-123 increased insulin-induced relaxation only in aorta segments from untreated HFHS mice (Figure 6D; *p* < 0.05).

### 3.11. Gene Expression of Inflammatory, Oxidative-Related Markers, and Receptors of Vasoactive Substances in Arterial Tissue

In arterial tissue a significant upregulation in the mRNA levels of the pro-inflammatory markers IL-6 (*p* < 0.05), COX-2 (*p* < 0.01), MCP-1 (*p* < 0.05), as well as the pro-fibrotic marker TGF-β (*p* < 0.01) were found in response to the HFHS diet, whereas the gene expression of IL-1β, TNF-α, and iNOS was unchanged (Figure 7A). CSAT+^®^ supplementation to HFHS-fed mice significantly reduced the MetS-induced overexpression of IL-6 (*p* < 0.01), COX-2 (*p* < 0.05), MCP-1 (*p* < 0.05), and TGF-β (*p* < 0.01).

In addition, MetS was associated with an overexpression of NOX-4 (*p* < 0.001), SOD-1 (*p* < 0.05), and GSR (*p* < 0.001) that was significantly prevented by CSAT+^®^ treatment (*p* < 0.001 for all). On the contrary, the mRNA levels of LO remained unchanged among experimental groups (Figure 7B).

Finally, obese mice showed an increased gene expression of α_1_-adrenoreceptor (*p* < 0.05), endothelin receptor type A (ETA) (*p* < 0.01), endothelin receptor type B (ETB) (*p* < 0.05), endothelin-1 (ET-1) (*p* < 0.01), and angiotensin receptor type 1 (AT-1) (*p* < 0.05) that was totally prevented by CSAT+^®^ supplementation.

### 3.12. Effects of Carob Treatment on Cardiac Function after Coronary Ischemia-Reperfusion

The effects of treatment with CSAT+^®^ on coronary perfusion pressure, heart contractility (dp/dt), and heart rate before and after IR are shown in Figure 8A–C, respectively.

No significant differences in coronary perfusion pressure, dp/dt, and heart rate were found among experimental groups before IR. On the contrary, after IR a significant reduction in coronary perfusion pressure was found in hearts from chow (*p* < 0.001), untreated HFHS (*p* < 0.01), and HFHS+ CSAT+^®^ mice (*p* < 0.05). Likewise, IR induced a reduction on heart contractility in all experimental groups although this reduction was only statistically significant in hearts from untreated HFHS-fed mice compared to chow animals (*p* < 0.001). No significant changes were found in heart rate among experimental groups.

### 3.13. Effects of CSAT+^®^ Treatment on Cardiomyocyte Apoptosis and Gene Expression of Pro-Inflammatory and Oxidative Stress Related mMarkers in Myocardial Tissue

After IR, results from TUNEL immunoassay showed a significant increase in cardiomyocyte apoptosis in hearts from untreated HFHS mice compared to both chow and HFHS mice treated with CSAT+^®^ (Figure 8D and 8E; *p* < 0.05). Likewise, MetS was associated with an up-regulation in the gene expression of IL-6, MCP-1, TGF-β, LO, GSR, and iNOS in myocardial tissue (Figure 8F; *p* < 0.05 for all) that was significantly attenuated by CSAT+^®^ supplementation (Figure 8F; *p* < 0.05 for all).

## 4. Discussion

In this manuscript, we show the beneficial effects of CSAT+^®^, a proprietary blend of carob (*Ceratonia silique* L.) pods and seeds extract together with FOS, improving the metabolic and cardiovascular alterations associated to MetS in mice.

In vitro, CSAT+^®^ exerts antioxidant capacity that could be due to the presence of several antioxidant compounds such as gallic, gallotannins, or phenolic compounds [43,44].

In vivo, our results show that supplementation with CSAT+^®^ does not affect body weight gain or adiposity in mice fed a HFHS diet. Likewise, other authors have reported that treatment with carob pod affects neither body weight, nor fat mass in vivo [27]. However, it exerts a strong delipidating effect in vitro on adipocyte cultures [31] and in vivo reduces liver triacylglycerol levels in obese rats [45] and in rabbits with hypercholesterolemia [28].

An important finding is that supplementation with CSAT+^®^ decreases the glycemic status and the HOMA-IR, suggesting improved insulin sensitivity. These results are in agreement with previous studies that report increased insulin sensitivity in response to carob treatment both in experimental animals [20] and in humans [18,19,46]. However, other authors have reported that carob consumption in combination with a glucose load increases plasma glucose and serum insulin responses in humans [47]. In diabetic rats, the insulin sensitizing effect is mediated by a decrease in the activity of the α-amylase and α-glucosidase and through a lower destruction of pancreatic β-cells that leads to improved insulin synthesis and secretion [20]. In our model, the beneficial effect on the glycemic status seems to be mediated by a reduction in MetS-induced insulin resistance in skeletal muscle and visceral adipose tissue, since we found a significant activation of the PI3K/Akt pathway in response to insulin in both visceral adipose tissue and gastrocnemius muscle explants from chow and CSAT+^®^ -treated mice but not in explants from mice fed with the HFHS diet alone. In visceral adipose tissue, the improved insulin resistance was associated with a decreased expression of pro-inflammatory markers such as IL-1β, TNF-α, and MCP-1 and with a decreased macrophage infiltration. Likewise, other authors have reported a direct anti-inflammatory effect of carob pod in LPS-stimulated macrophages [27,31]. Like in AT, we also found a lower gene expression of MCP-1 and decreased macrophage infiltration in gastrocnemius explants from CSAT+^®^-supplemented mice, although the mRNA levels of pro-inflammatory cytokines were unchanged among experimental groups. However, we found a significant increase in the gene expression of the antioxidant enzymes SOD-1 and GSR which is in agreement with the antioxidant capacity shown in previous studies with carob pod extracts [31] and with extracts from other parts of carob fruit and tree [48].

In addition to the beneficial effects in the glycemic status and MetS-induced insulin resistance, supplementation with CSAT+^®^ also improved the lipid profile by decreasing the circulating concentrations of both total and LDL-cholesterol. Our results are in agreement with those reported by other authors both in experimental animals [22,23] and in humans [25,26,49]. However, contrary to other studies [45,50,51], CSAT+^®^ treatment was unable to reduce triglyceride levels neither in hepatic tissue (data not shown), nor in the plasma. The discrepancies between these studies may be explained by differences in the species used (mice vs rats and rabbits) or by differences in the duration of the high fat diet. In our study, diet was extended over 26 weeks, whereas in the other studies, it had a duration of only eight weeks. Indeed, we analyzed total cholesterol, LDL-c, and TG plasma levels after 12 weeks of diet and we found a significant reduction in all of them in obese mice treated with CSAT+^®^ compared to untreated mice (data not shown). Thus, it is possible that although the effects of carob treatment on cholesterol are maintained over time, the effects of carob treatment on TG disappear after a long-time consumption of HFHS diet. Differences could also be explained by the dose of carob extract. In this study, the dose of carob extract was similar to the one used in previous studies with rodents [23,45] and higher than the one used in human studies [47,49,52]. New experiments are needed to assess the specific dose of this specific extract for human consumption.

Our results show not only a positive effect of CSAT+^®^ decreasing MetS-induced hypertension, data from vascular reactivity and reactive hyperemia experiments show that the significant reduction in mean blood pressure seems to be the result of improved endothelial function, as it is demonstrated by the increased relaxation of aortic rings in response to Ach and by the improved blood flow after occlusion of femoral artery. In addition, the decreased blood pressure in CSAT+^®^-treated mice may also be explained by the decreased vascular contraction of aorta segments in response to the vasoconstrictor AngII. Likewise, other authors have reported a beneficial effect of a carob pod extract in vascular function in hypercholesterolemic rabbits that is demonstrated by an increased relaxation of aorta segments to Ach and by a decreased response to the vasoconstrictor KCl [28]. Like in the study by Valero et al. we also found a significant reduction in the mRNA levels of several pro-inflammatory cytokines and a decreased expression of the pro-fibrotic marker TGF-1β in arterial tissue, which means that the improved vascular function is associated with decreased vascular fibrosis and inflammation. In addition, a novelty of our study is that supplementation with CSAT+^®^ also prevents the MetS-induced overexpression of several enzymes involved in oxidative stress. These results are important since, to our knowledge, this is the first study that demonstrates a direct antioxidant effect of carob in arterial tissue in vivo. Interestingly, treatment with the carob extract not only reduces the gene expression of the pro-oxidant enzyme NOX-1 in arterial tissue but also the gene expression of the anti-oxidant enzymes SOD-1 and GSR which possibly denotes a lower oxidative status that requires lower activity of antioxidant enzymes. This fact was further demonstrated by the results obtained from the dose-response curves to Ach in the presence/absence of antioxidants. In these experiments, we observed that pre-incubation of aorta segments with antioxidants improved the vascular relaxation in response to Ach in untreated obese mice but not in aorta segments from lean and obese mice supplemented with CSAT+^®^. Thus, these results clearly denote a higher oxidative status in aorta segments from untreated obese mice compared to both lean and CSAT+^®^-treated animals that compromises endothelial function, as it has been previously reported by other authors [53,54,55].

Another important finding of our work is that, supplementation with CSAT+^®^ attenuates the MetS-induced insulin vascular resistance. This effect was demonstrated by a higher activation of the PI3K/Akt pathway in arterial tissue from CSAT+^®^-treated mice in response to insulin that leads to improved relaxation of aorta segments compared to segments from obese untreated mice. In addition, since insulin is reported to induce vascular relaxation through different pathways depending on the context, we also examined the mechanisms by which insulin exerts vasodilation in aorta segments of mice from the three experimental groups. These mechanisms include not only the release of nitric oxide by vascular endothelial cells [15], but also the release of other vasoactive substances, such as endothelin-1 or prostaglandins [56]. On one hand, our results reveal that insulin-induced vasodilation is mediated by endothelial NO release in aorta segments from all experimental groups since vascular relaxation is reduced in the presence of the unspecific nitric oxide synthase blocker L-NAME. On the other hand, our results also show that decreased vascular insulin sensitivity was produced by different mediators in obese mice untreated or supplemented with CSAT+^®^. In untreated obese mice, vascular insulin resistance seemed to be mediated by ET-1, since insulin-induced vasodilation was improved in the presence of the ET-A blocker BQ-123. However, in CSAT+^®^-treated mice, the partial vascular insulin resistance seemed to be mediated by prostanoids as insulin-induced vasodilation was improved in the presence of the cyclooxygenase blocker indomethacin.

Finally, one of the most relevant findings of our study is that supplementation with CSAT+^®^ to obese mice prevents the cardiac damage after coronary ischemia-reperfusion. This cardioprotective effect is shown by the improved heart contractility of hearts from CSAT+^®^-treated mice compared to untreated mice with MetS after IR, and seems to be mediated by decreased expression of markers related to both, inflammation and oxidative stress that lead to decreased apoptosis of cardiomyocytes. Although other studies had demonstrated a beneficial effect of carob on morphological cardiac parameters in hypercholesterolemic animals ^22^, to our knowledge, this is the first study reporting a cardioprotective effect of carob pod extract in terms of cardiac function after an acute myocardial infarct.

In conclusion, supplementation with a carob pod and seed extract enriched with FOS improves the lipid profile, exerts cardioprotective effects, and attenuates metabolic and vascular insulin resistance in obese mice. Thus, it may constitute an interesting dietary intervention to treat/prevent the cardiometabolic alterations associated with MetS.

## Figures and Tables

**Figure 1 antioxidants-09-00339-f001:**
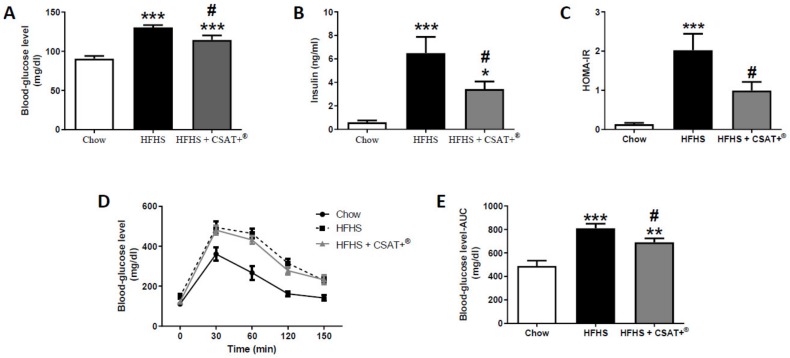
Blood glucose (**A**) and insulin (**B**) plasma levels, Homeostatic Model Assessment of Insulin Resistance (HOMA-IR) (**C**), oral glucose tolerance test (**D**) and area under the curve (**E**) in mice fed a standard diet (Chow), a high fat diet/sucrose diet (HFHS) or high fat diet/sucrose diet supplemented with the carob pod extract (HFHS + CSAT+^®^). * *p* < 0.05; ** *p* < 0.01; *** *p* < 0.001 vs. chow. # *p* < 0.05 vs. HFHS. Values are represented as mean ± S.E.M; *n* = 12 samples/experimental group.

**Figure 2 antioxidants-09-00339-f002:**
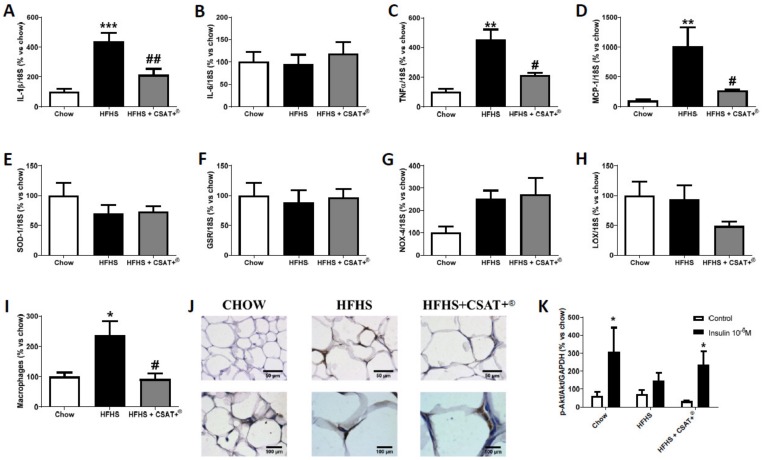
mRNA levels of interleukin-1 beta (**A**), interleukin-6 (**B**), tumor necrosis factor-alpha (**C**), monocyte chemoattractant protein (**D**), superoxide dismutase-1 (**E**), glutathione reductase (**F**), NADPH oxidase-4 (**G**), and lipoxygenase (**H**) in retroperitoneal adipose tissue of mice fed a standard diet (Chow), a high fat diet/sucrose diet (HFHS) or high fat diet/sucrose diet supplemented with the carob pod extract (HFHS + CSAT+^®^). Macrophages infiltration was measured by immunohistochemistry (**I**,**J**) and pAKT/AKT ratio (**K**) was measured in retroperitoneal adipose tissue after 15 min of explant incubation with or without 10^−6^ M insulin. * *p* < 0.05; ** *p* < 0.01; *** *p* < 0.001 vs. chow # *p* < 0.05; ## *p* < 0.01 vs. HFHS. Values are represented as mean ± S.E.M; *n* = 5–8 samples/experimental group.

**Figure 3 antioxidants-09-00339-f003:**
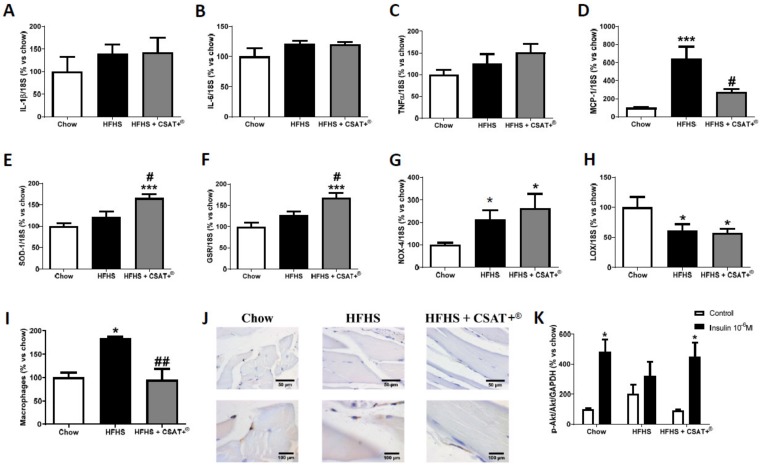
mRNA levels of interleukin-1 beta (**A**), interleukin-6 (**B**), tumor necrosis factor-alpha (**C**), monocyte chemoattractant protein (**D**), superoxide dismutase-1 (**E**), glutathione reductase (**F**), NADPH oxidase-4 (**G**), and lipoxygenase (**H**) in gastrocnemius muscle of mice fed a standard diet (Chow), a high fat diet/sucrose diet (HFHS), or high fat diet/sucrose diet supplemented with the carob pod extract (HFHS + CSAT+^®^). Macrophages infiltration was measured by immunohistochemistry (**I**,**J**) and pAKT/AKT ratio (**K**) was measured in this gastrocnemius explants after 15 min of incubation with or without 10^−6^ M insulin. * *p* < 0.05; *** *p* < 0.001 vs. chow. # *p* < 0.05; ## *p* < 0.01 vs. HFHS. Values are represented as mean ± S.E.M; *n* = 5–8 samples/experimental group.

**Figure 4 antioxidants-09-00339-f004:**
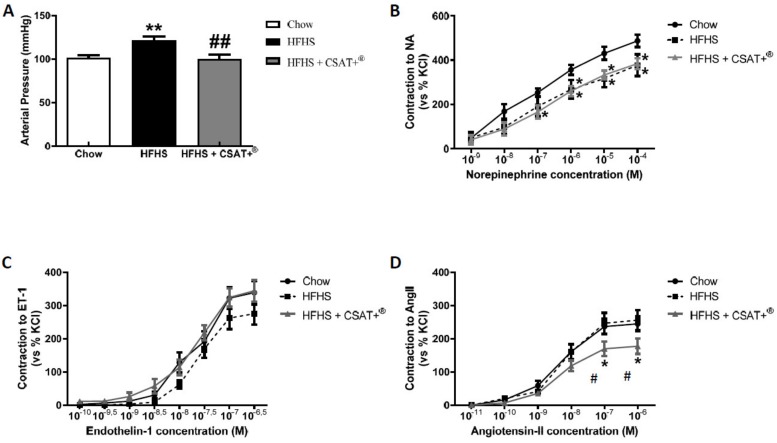
Mean arterial pressure (**A**) and vascular response of abdominal aortic rings to cumulative dose-response of norepinephrine (10^−9^ to 10^−4^ M) (**B**), endothelin-1 (10^−10^ to 10^−6.5^ M) (**C**) angiotensin-II (10^−11^ to 10^−6^ M) (**D**) of mice fed a standard diet (Chow), a high fat diet/sucrose diet (HFHS) or high fat diet/sucrose diet supplemented with the carob pod extract (HFHS + CSAT+^®^). * *p* < 0.05; ** *p* < 0.01 vs. chow. # *p* < 0.05; ## *p* < 0.01 difference vs. HFHS. Values are represented as mean ± S.E.M; *n* = 6–8 samples ring/experimental group.

**Figure 5 antioxidants-09-00339-f005:**
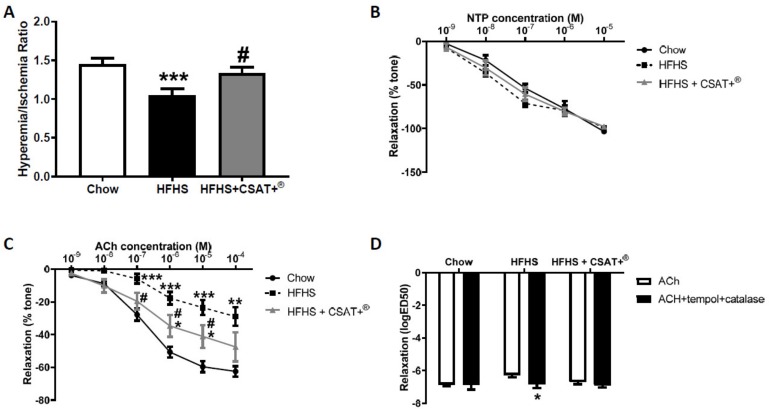
Reactive hyperemia (**A**),relaxation of thoracic aortic rings to sodium nitroprusside (**B**) and acetylcholine alone (**C**) or in the in the presence/absence of tempol+catalase (**D**) of mice fed a standard diet (Chow), a high fat diet/sucrose diet (HFHS), or high fat diet/sucrose diet supplemented with the carob pod extract (HFHS + CSAT+^®^). * *p* < 0.05; ** *p* < 0.01 vs. chow; *** *p* < 0.001 vs. chow. # *p* < 0.05 vs. HFHS. Values are represented as mean ± S.E.M; *n* = 10–12 samples/experimental group.

**Figure 6 antioxidants-09-00339-f006:**
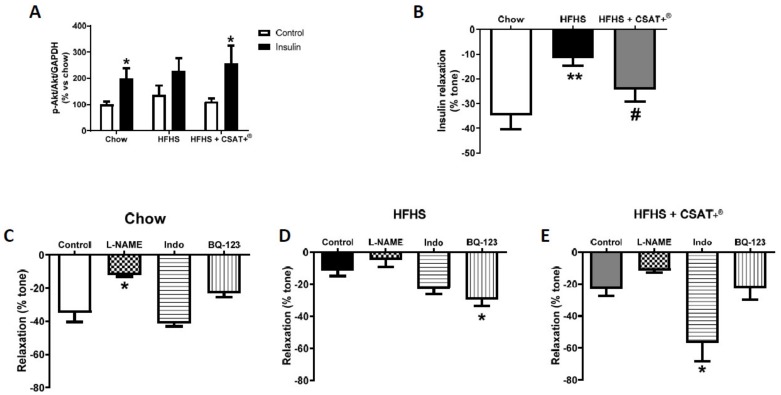
pAKT/AKT ratio in aortic tissue incubated 15 min with or without insulin 10^−6^ M (**A**) and relaxation of thoracic aortic rings in response to insulin alone (**B**) or in the presence of Nω-nitro-L-arginine methyl ester hydrochloride (L-NAME) (**C**), indomethacin (**D**), or BQ-123 (**D**) from mice fed a standard diet (Chow), a high fat diet/sucrose diet (HFHS), or high fat diet/sucrose diet supplemented with the carob pod extract (HFHS + CSAT+^®^). * *p* < 0.05; ** *p* < 0.01 vs. chow. # *p* < 0.05 vs. HFHS. Values are represented as mean ± S.E.M; *n* = 6–10 aortic rings/experimental group.

**Figure 7 antioxidants-09-00339-f007:**
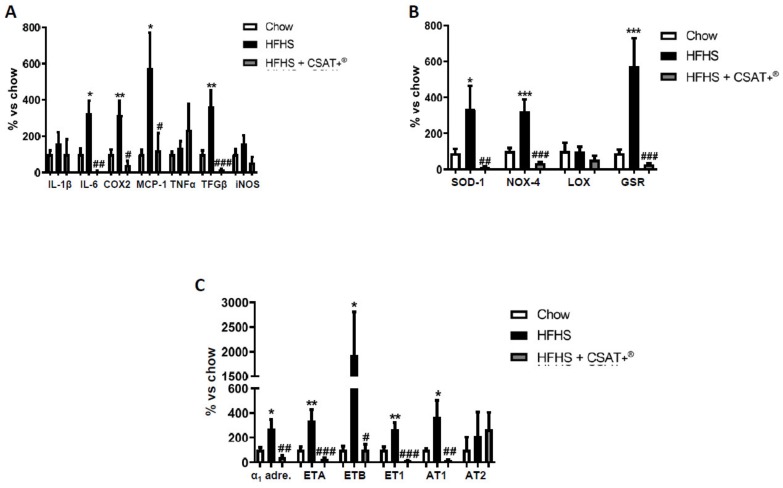
mRNA levels of interleukin-1 beta (IL-1β), interleukin-6 (IL-6), cyclooxygenase 2 (COX2), monocyte chemoattractant protein (MCP-1), tumor necrosis factor-alpha (TNFα), transforming growth factor beta (TGFβ) and inducible nitric oxide synthase (iNOS) (**A**), superoxide dismutase-1 (SOD-1), NADPH oxidase-4 (NOX-4), lysyl oxidase (LOX) and glutathione reductase (GSR) (**B**) and alpha-1 adrenergic receptor (α_1_ adre), endothelin receptor type A (ETA), endothelin receptor type B (ETB), endothelin-1 (ET-1), angiotensin receptor type 1 (AT1) and angiotensin receptor type 2 (**C**) in aortic tissue of mice fed a standard diet (Chow), a high fat diet/sucrose diet (HFHS), or high fat diet/sucrose diet supplemented with the carob pod extract (HFHS + CSAT+^®^). * *p* < 0.05; ** *p* < 0.01; *** *p* < 0.001 vs. chow. # *p* < 0.05; ## *p* < 0.01; ### *p* < 0.01 vs. HFHS. Values are represented as mean ± S.E.M; *n* = 8 samples/experimental group.

**Figure 8 antioxidants-09-00339-f008:**
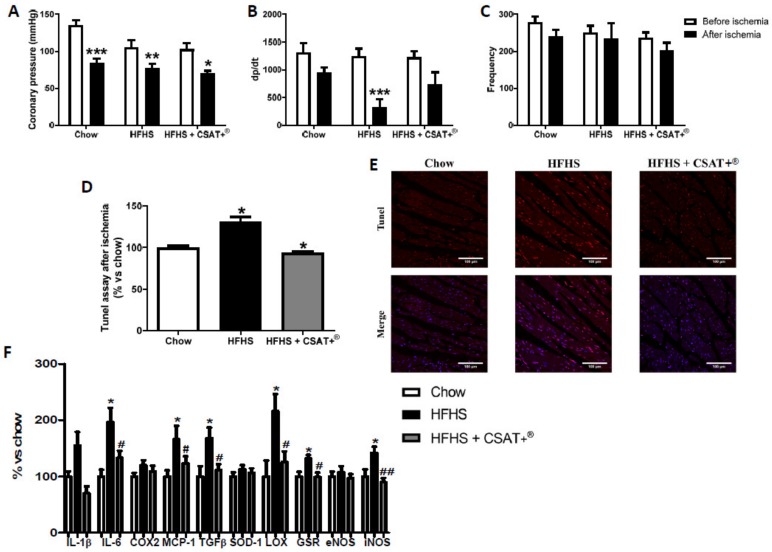
Changes in coronary perfusion pressure (**A**), dP/dt (**B**) and heart rate (**C**) before and after coronary ischemia-reperfusion in hearts from mice fed a standard diet (Chow), a high fat diet/sucrose diet (HFHS), or high fat diet/sucrose diet supplemented with the carob pod extract (HFHS+CSAT+^®^). Apoptosis of ischemic hearts were analyzed by TUNEL assay (**D**,**E**); cardiac mRNA levels (**F**) of interleukin-1 beta, interleukin-6, cyclooxygenase 2, monocyte chemoattractant protein, transforming growth factor beta, superoxide dismutase-1, lysyl oxidase, glutathione reductase, endothelial nitric oxide synthase and inducible nitric oxide synthase in ischemic hearts. * *p* < 0.05; ** *p* < 0.01; *** *p* < 0.001 vs. chow and # *p* < 0.05 vs. HFHS; ## *p* < 0.01 vs. HFHS. Values are represented as mean ± S.E.M; *n* = 6–8 samples/experimental group.

**Table 1 antioxidants-09-00339-t001:** Ultraviolet–visible (UV-VIS) absorption and mass spectroscopic data (negative ionization mode; [M–H]^−^) of the identified polar phenolic compounds found in CSAT+^®^ by high performance liquid chromatography mass Spectrometry (HPLC-MS). T_R_ = Retention time; sh = shoulder.

T_R_ (min)	Molecule	UV-VIS (nm)	[M–H]^−^ (*m*/*z*)	Fragment Ions (*m*/*z*)
4.2	Gallic acid	229/274	169.0	125.1
4.5	Monogalloyl	227/278	331.0	169.1
	hexoside			
4.8	Monogalloyl	226/278	331.0	168.9
	hexoside			
5.3	Trigalloyl hexoside	226/276	635.0	483.0, 331.0, 313.0,
				271.0, 211.0, 169.1
8.0	Digalloyl hexoside	227/275	483.0	331.0, 313.0, 169.0
9.5	Digalloyl hexoside	226/275	483.0	443.0, 331.0, 313.0,
				210.9, 169.0, 124.1
10.7	Digalloyl hexoside	226/277	483.0	331.0, 313.0, 169.0
11.5	Digalloyl hexoside	228/280	483.0	331.1
13.1	Trigalloyl hexoside	224/273	635.0	483.0, 464.7, 331.0,
				312.9, 271.0, 210.9,
				168.9
15.1	Trigalloyl hexoside +	267	635.0	483.0, 331.0, 312.9,
	unknown		443.0	270.8, 210.9, 168.9
15.8	Trigalloyl hexoside	280	634.8	482.9, 331.0, 313.0,
				271.0, 241.1, 211.1,
				168.9
18.2	Trigalloyl hexoside	224/272	635.0	483.0, 464.9, 331.0,
				313.0, 271.0, 241.1,
				211.1, 168.9
19.6	Trigalloyl hexoside	224/272	635.8	483.0, 464.9, 422.9,
				331.0, 313.0, 270.9,
				241.0, 210.9, 193.1,
				168.9
21.0	Trigalloyl hexoside	224/272	635.1	483.0, 464.9, 422.9,
				331.0, 313.0, 271.0,
				240.8, 210.9, 193.1,
				168.9
21.6	Trigalloyl hexoside	224/278	634.9	483.1, 465.0, 421.0,
				331.0, 313.0, 271.0
22.3	Tetragalloyl	223/272	786.9	635.0, 483.0, 330.8,
	hexoside			313.0, 271.0, 211.0,
				168.9
23.8	Trigalloyl hexoside	223/272	634.5	483.1, 465.0, 443.0,
				313.0, 168.9
24.9	Tetragalloyl	225/275	787.1	635.0, 483.1, 465.8,
	hexoside			443.0, 423.0, 313.0,
				271.1, 168.9, 151.1
27.0	Tetragalloyl	224/278	786.9	635.0, 483.0, 465.8,
	hexoside			449.0, 313.0, 271.0,
				210.7, 168.9, 121.1
28.2	Trigalloyl hexoside	226/285	635.0	483.1, 465.8, 449.0,
				401.1, 313.0, 271.0,
				210.7
29.9	Trialloyl hexoside	225/276	634.9	482.8, 465.0, 313.0
31.1	Tetragalloyl	223/271	787.0	635.0, 482.9, 464.9,
	hexoside			168.9
32.3	Tetragalloyl	223/274	787.0	635.0, 482.9, 464.9,
	hexoside			168.9
35.2	Tetragalloyl	223/270	786.8	635.0, 465.8, 403.0,
	hexoside			124.1
37.2	Tetragalloyl	263/278/370	786.9	635.0, 464.9, 317.0
	hexoside + Myricetin		479.0	
	hexoside			
40.0	Tetragalloyl	223/278	786.9	635.0, 401.0, 313.0,
	hexoside			210.9, 169.0
45.1	Tetragalloyl	225/276	787.0	635.0, 617.0, 465.8
	hexoside			
47.3	Tetragalloyl	223/281/393sh	786.9	635.0, 616.9, 462.8,
	hexoside			271.0, 124.8
49.5	Tetragalloyl	223/277/355	787.0	617.0, 464.9, 271.0,
	hexoside + Myricetin			168.8,
	deoxy-hexoside (1)		463.0	271.0, 316.0
51.3	Myricetin deoxy-	260/300sh/348	463.0	316.9, 315.9, 287.0,
	hexoside (2)			271.0
52.1	Ellagic acid hexoside	250/369	463.0	301.0, 229.0, 201.1,
				169.1
54.2	Quercetin hexoside	253/298sh/352	462.9	300.9, 270.8, 254.9,
				151.1
56.5	Naringin	225/281/335	579.0	459.1, 271.0
58.6	Isohesperetin	229/281/328	579.0	459.1, 2710
	(Narirutin)			
59.8	Quercetin deoxy-	226/253/261sh/3	447.0	300.0, 301.0, 271.0,
	hexoside	48		255.0
62.2	Hesperetin	226/283/331	609.1	461.1, 446.9, 300.9
	rutinoside			
71.0	Kaempferol	221/251/263sh/3	593.1	285.0
	rutinoside	45		
72.7	Methylquercetin	220/253/294sh/3	314.9	299.9, 271.0
		54		

**Table 2 antioxidants-09-00339-t002:** Body weight, daily food intake and weights of retroperitoneal visceral adipose tissue, lumbar subcutaneous adipose tissue, interscapular brown adipose tissue, heart, kidneys, adrenal glands, spleen, liver, gastrocnemius, and soleus of mice fed either a standard diet (chow) or a high fat/high sucrose diet (HFHS) supplemented or not with carob fruit extract (CSAT+^®^) for 26 weeks. Data are represented as mean ± SEM. (*n* = 12 mice/group). * *p* < 0.05 vs. chow; ** *p* < 0.01 vs. chow; *** *p* < 0.001 vs. chow; # *p* < 0.05 vs. HFHS; ## *p* < 0.01 vs. HFHS.

	Chow	HFHS	HFHS + CSAT+^®^
Body weight (g)	29.7 ± 0.54	46.4 ± 1.6 ***	48.7 ± 1.3 ***
Food intake (g/mouse/day)	3.8 ± 0.03	2.8 ± 0.02 ***	2.9 ± 0.03 ***
Visceral Retroperitoneal adipose tissue	372 ± 35.3	1556 ± 67.3 ***	1414 ± 79.2 ***
(mg/cm)			
Subcutaneous lumbar adipose tissue (mg/cm)	125.6 ± 8.5	945 ± 76.3 ***	873 ± 93.2 ***
Interescapular Brown adipose tissue (mg/cm)	67.2 ± 5.5	122 ± 13.7 ***	174 ± 11.8 *** ##
Heart (mg/cm)	107.6 ± 4.4	130 ± 4.6 **	113 ± 4.1 #
Kidneys (mg/cm)	205.9 ± 4.9	257 ± 12.4 **	234 ± 9.1
Adrenal glands (mg/cm)	1.3 ± 0.1	1.80 ± 0.15 *	1.75 ± 0.11 *
Spleen (mg/cm)	39.9 ± 2.6	51.4 ± 4.5 **	48.7 ± 2.7 **
Liver (mg/cm)	620 ± 20.5	896 ± 62.3 *	931 ± 74.6 **
Gastrocnemious (mg/cm)	78.5 ± 2.6	83.2 ± 2.5	80.3 ± 2.4
Soleus (mg/cm)	4.9 ± 0.2	7.3 ± 0.6 **	5.7 ± 0.3 #

**Table 3 antioxidants-09-00339-t003:** Plasma levels of adiponectin, leptin, total lipids, triglycerides, total cholesterol, low-density lipoproteins (LDL)-cholesterol, high-density lipoproteins (HDL)-cholesterol, and interleukin-6 (IL-6) of mice fed either a standard diet (chow) or a high fat/high sucrose diet (HFHS) supplemented or not with CSAT+^®^ for 26 weeks. Data are represented as mean ± SEM. (*n* = 12 mice/group). * *p* < 0.05 vs. chow; ** *p* < 0.01 vs. chow; *** *p* < 0.001 vs. chow; # *p* < 0.05 vs. HFHS.

	Chow	HFHS	HFHS + CSAT+^®^
Adiponectin (ng/mL)	4771 ± 598	3084 ± 443.8 *	5805 ± 828 #
Leptin (ng/mL)	1.06 ± 0.3	22.5 ± 2.8 ***	26.5 ± 1.9 ***
Total lipids (mg/dL)	3752 ± 150	5011 ± 274 ***	5284 ± 198 ***
Triglycerides (mg/dL)	68.9 ± 4.3	124 ± 9.1 ***	105 ± 4.8 ***
Total cholesterol (mg/dL)	175 ± 9.8	390 ± 11.6 ***	339 ± 18.6 *** #
LDL-c (mg/dL)	76.1 ± 6.7	189.8 ± 9.9 ***	144 ± 12.9 *** #
HDL-c (mg/dL)	73.8 ± 10.4	197.7 ± 26.8 ***	181 ± 23.2 **
IL-6 (pg/mL)	39.1 ± 4.3	61.7 ± 4.6 *	46.1 ± 2.8 #

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
