# Peer review of "Supplementation with a Carob (Ceratonia siliqua L.) Fruit Extract Attenuates the Cardiometabolic Alterations Associated with Metabolic Syndrome in Mice"

_antioxidants, 2020, doi:10.3390/antiox9040339_

Round 1
Reviewer 1 Report
The current manuscript presents an impressive series of experiments that assess the effectiveness of 26 weeks of proprietary carob pod and seed extract supplementation on metabolism and vascular reactivity in a mouse model of metabolic syndrome (MetS). The data presented are of interest, and deserving of publication, however, the manuscript as it stands needs revision.
Some suggestions are provided below:
Please limit the use of 1-2 sentence “paragraphs”. In both the introduction and discussion there are a number of these “strands of thought”, please rework these into either fully supported paragraphs or consider eliminating them if they are not key to the discussion.
Methods:
Line 100: Please fix the typo.
Line 149: Why 4.8% CSAT? That is a rather specific dose. Also, some discussion somewhere in the manuscript as to how this supplementation level will related back to human intake would be of value.
General: Since you are anesthetizing the animals euthanize or killed is preferred over sacrifice, which usually denotes something done for the gods.
Also, may want to reorganize the methods so that the characterization, metabolic inflammation/oxidative stress, and cardiovascular measures are grouped together. For example, sections 2.3.3, 2.3.4, 2.3.9, and possibly 2.3.7 could follow each other sequentially as the “cardiovascular” measures.
Line 182: please further define “aorta” as either thoracic or abdominal.
Line 190: I believe you mean “prior to the addition of” instead of “to subsequently add”
Section 2.3.4 Please add the methods for the vasoconstriction assays.
Line 196: You may want to structure the beginning of the sentence as, “ In order to study the mechanism of insulin response, some segments….”
Section 23.3.12
Please add how the between group analysis was conducted in the dose-response aortic ring experiments. At the minimum it should be a two-way ANOVA with post hoc.
Results:
Major comment: You have a lot of results/experiments here, so similar to above, you may want to consider organizing it so that your section subheadings actually describe the effects observed in a logical, almost story telling fashion, rather than systematically stepping through the experiments and having the reader work at keeping track of all of the observed effects, and how they are related to one another.
Other minor comments for the results:
Line 326: Best not to use “among these that stand out” as it suggests that you will further describe why they are important. Instead I suggest using something along the lines of “These include” with a little more detail as to what class/type of phenolic they represent. If you have any data along the lines with regards to the relative amounts of these phenolics or what classes dominate that would be desired.
Line 331-332: Without any comparator with a known antioxidant, it is hard to interpret the total antioxidant capacity measurement. I would suggest eliminating it or repeat the experiment with a known antioxidant as a positive control.
Lines 430-431: Details of blood pressure measurements are not described in the methods. How were the blood pressure measurements obtained?
Discussion:
The major issue with the discussion is that it does not succinctly summarize and discuss the findings, particularly with regards to the vascular findings.
In addition, please go through and moderate some of the definitive statements, such as:
Line 550-551: Without actual quantification of the compounds detected, along with no positive control for the antioxidant capacity measurement, I would avoid such a definitive statement. Also, you have other flavonoids, polyphenols in your characterization that are also known antioxidants.
Line 552: As there was not a CSAT+ alone group, you do not know its true effect on weight, only that it did not worsen or improved adiposity and weight gain observed in the MetS model.
Author Response
Reviewer 1
The current manuscript presents an impressive series of experiments that assess the effectiveness of 26 weeks of proprietary carob pod and seed extract supplementation on metabolism and vascular reactivity in a mouse model of metabolic syndrome (MetS). The data presented are of interest, and deserving of publication, however, the manuscript as it stands needs revision.
Some suggestions are provided below:
Please limit the use of 1-2 sentence “paragraphs”. In both the introduction and discussion there are a number of these “strands of thought”, please rework these into either fully supported paragraphs or consider eliminating them if they are not key to the discussion.
The introduction and the discussion sections have been carefully revised in order to rewrite or delete such paragraphs. However, as the reviewer did not mention which specific paragraphs needed revision it is possible that some of them have not been corrected.
Methods:
Line 100: Please fix the typo.
It has been corrected
Line 149: Why 4.8% CSAT? That is a rather specific dose. Also, some discussion somewhere in the manuscript as to how this supplementation level will related back to human intake would be of value.
The dose was selected taking into account the normal amount of dietary fiber in rodent diets which is between 5-7%. In this customized diet 5% cellulose, which is the most common type of insoluble fiber present in rodent diets, was substituted by 4,8% CSAT+ which is composed mainly by fiber (70-80%).
Obese mice ate approximately 3g of chow/day, so the amount of CSAT+ that they ingested per day was 144mg. Taking into account that they had a medium weight of 40g the average dose per day was approximately 3,6 mg/g. This dose was similar to the one used by other authors in experiments with rodents:
- Macho-González et. al 2019: 4g/kg
- Rico et. al 2019: 5%
Regarding the relationship with human intake, humans are recommended to ingest 40g fiber/day. For a person of 60kg that would be 0,66g/kg, which is approximately 5 times lower than the amount administered to the mice used in this study. However, in experiments with experimental animals the doses of treatments are normally higher than the ones administered to humans.
- Zunft et. al.:15 g/day
- Gruendel et. al. 2006: 20 g/day
- Gruendel et. al. 2007: 50 g/day
As suggested by the reviewer, the following sentence regarding the CSAT+ dose has been included in the Discussion section (Lines 600-603):
“Differences could also be explained by the dose of carob extract. In this study, the dose of carob extract was similar to the one used in previous studies with rodents 23, 45 and higher than the one used in human studies 47, 49, 52. New experiments are needed to assess the specific dose of this specific extract for human consumption.”
References:
- Macho-González A, Garcimartín A, López-Oliva ME, Ruiz-Roso B, Martín de la Torre I, Bastida S, Benedí J, Sánchez-Muniz FJ. Can Carob-Fruit-Extract-Enriched Meat Improve the Lipoprotein Profile, VLDL-Oxidation, and LDL Receptor Levels Induced by an Atherogenic Diet in STZ-NAD-Diabetic Rats? 2019 Feb 3;11(2). pii: E332. doi: 10.3390/nu11020332.
- Rico D1, Martin-Diana AB2, Lasa A3,4, Aguirre L5,6, Milton-Laskibar I7,8, de Luis DA9, Miranda J10,11.Effect of Wakame and Carob Pod Snacks on Non-Alcoholic Fatty Liver Disease. Nutrients. 2019 Jan 4;11(1). pii: E86. doi: 10.3390/nu11010086.
- Zunft H.J.F., Lüder W., Harde A., Haber B., Graubaum H.J., Koebnick C., Grünwald J. 2003. Carob pulp preparation rich in insoluble fibre lowers total and LDL cholesterol in hypercholesterolemic patients. European Journal of Nutrition, 42, 235-242
- Gruendel S., García A.L., Otto B., Mueller C., Steiniger J., Weickert M.O., Speth M., Katz N., Koebnick C. 2006. Carob Pulp Preparation Rich in Insoluble Dietary Fiber and Polyphenols Enhances Lipid Oxidation and Lowers Postprandial Acylated Ghrelin in Humans. The Journal of Nutrition, 136(6), 1533-1538
- Gruendel S., García A.L., Otto B., Wagner K., Bidlingmaier M., Burget L., Weickert M.O., Dongowski G., Speth M., Katz , Koebnick C. 2007. Increased acylated plasma ghrelin, but improved lipid profiles 24-h after onsumption of carob pulp preparation rich in dietary fibre and polyphenols. British Journal of Nutrition, 98(6), 1170-1177
General: Since you are anesthetizing the animals euthanize or killed is preferred over sacrifice, which usually denotes something done for the gods.
This has been corrected as suggested.
Also, may want to reorganize the methods so that the characterization, metabolic inflammation/oxidative stress, and cardiovascular measures are grouped together. For example, sections 2.3.3, 2.3.4, 2.3.9, and possibly 2.3.7 could follow each other sequentially as the “cardiovascular” measures.
The material and methods section has been reorganized as suggested by the reviewer grouping the metabolic and the cardiovascular procedures.
Line 182: please further define “aorta” as either thoracic or abdominal.
The following sentence has been inserted in the material and methods section (Lines 205-206): “Thoracic segments were used for the vasodilation studies and abdominal segments for the vasoconstriction studies.”
Line 190: I believe you mean “prior to the addition of” instead of “to subsequently add”
It has been corrected
Section 2.3.4 Please add the methods for the vasoconstriction assays.
Methods for the vasoconstriction studies are now included in the material and methods section (Lines 227-230). We thank the reviewer for this important observation.
Line 196: You may want to structure the beginning of the sentence as, “ In order to study the mechanism of insulin response, some segments….”
It has been corrected as suggested
Section 23.3.12
Please add how the between group analysis was conducted in the dose-response aortic ring experiments. At the minimum it should be a two-way ANOVA with post hoc.
The statistical analysis of the dose-response aortic ring experiments was performed by two-way ANOVA followed by Bonferroni post-hoc test. It is now stated in the Material and Methods section
Results:
Major comment: You have a lot of results/experiments here, so similar to above, you may want to consider organizing it so that your section subheadings actually describe the effects observed in a logical, almost story telling fashion, rather than systematically stepping through the experiments and having the reader work at keeping track of all of the observed effects, and how they are related to one another.
We appreciate the reviewer’s comment. However, we consider that results are written in a coherent manner, describing the metabolic effects first and the cardiovascular effects secondly as suggested by the reviewer.
Other minor comments for the results:
Line 326: Best not to use “among these that stand out” as it suggests that you will further describe why they are important. Instead I suggest using something along the lines of “These include” with a little more detail as to what class/type of phenolic they represent.
It has been corrected as suggested.
If you have any data along the lines with regards to the relative amounts of these phenolics or what classes dominate that would be desired.
Unfortunately, we don’t have the data of the relative amounts of these phenolic compounds or what classes dominate. Due to the actual situation in our country for the coronavirus pandemic, our lab is now closed and we are unable to perform those experiments in the short term.
Line 331-332: Without any comparator with a known antioxidant, it is hard to interpret the total antioxidant capacity measurement. I would suggest eliminating it or repeat the experiment with a known antioxidant as a positive control.
We could eliminate the results of the antioxidant activity in vitro if the reviewer considers that it is better for the quality of the manuscript. However, we think that the Trolox assay is an accepted and well-characterized method to measure the antioxidant capacity of botanical extracts (MacDonald-Wicks et. al 2006). Indeed, other authors compare the antioxidant activity of different vegetal compounds (Bohm et. al 2002) or foods (Floegel et. al 2011) using this method.
The Trolox assay was first described by Miller et. al in 1993 and improved by Re et. al in 1999 and is based in the comparison of the antioxidant capacity of samples with an unknown antioxidant activity with the antioxidant activity of Trolox, an analogue of Vitamin E with strong antioxidant capacity. For this reason, we think that the method that we have used to measure the antioxidant capacity of CSAT+, already includes a comparison with a well-known antioxidant, as suggested by the reviewer.
References:
- MacDonald-Wicks L.K., Wood L.G., Garg M.L. 2006. Methodology for the determination of biological antioxidant capacity in vitro: a review. J Sci Food Agric, 86, 2046-2056
- Miller N.J., Rice-Evans C.A., Davies M.J., Gopinathan V., Milner A.A. 1993. A novel method for measuring antioxidant capacity and its application to monitoring antioxidant status in premature neonates. Clin Sci, 84, 407-412
- Re R., Pellegrini N., Proteggente A., Pannala A., Yang M., Rice-Evans C.A. 1999. Antioxidant activity applying an improved ABTS radical cation decolorization assay. Free Rad Biol Med, 26, 1231-1237
- Bohm V., Puspitasari-Nienaber, N.L., Ferruzzi M.G., Schwartz S.J. 2002. Trolox Equivalent Antioxidant Capacity of Different Geometrical Isomers of α-Carotene, β-Carotene, Lycopene, and Zeaxanthin. J Agric Food Chem, 50, 221-226
- Floegel A., Kim D.O., Chung S.J., Koo S.I., Chun O.K. 2011. Comparison of ABTS/DPPH assays to measure antioxidant capacity in popular antioxidant-rich US foods. J Food Comp Analysis, 24, 1043-1048
Lines 430-431: Details of blood pressure measurements are not described in the methods. How were the blood pressure measurements obtained?
The method for the measurement of blood pressure is now included in the Material and Methods section (Lines 182-191). We thank the reviewer for this important observation.
Discussion:
The major issue with the discussion is that it does not succinctly summarize and discuss the findings, particularly with regards to the vascular findings.
As stated above by the reviewer this manuscript contains many results and this makes difficult to discuss all of them succinctly. We have tried to discuss all the results in a coherent and organized manner, but we assume that it may result a little bit overwhelming for the reader due to the high number of data.
In addition, please go through and moderate some of the definitive statements, such as:
Line 550-551: Without actual quantification of the compounds detected, along with no positive control for the antioxidant capacity measurement, I would avoid such a definitive statement. Also, you have other flavonoids, polyphenols in your characterization that are also known antioxidants.
The sentence “In vitro, CSAT+® exerts antioxidant capacity that is most likely due to the presence of gallic acid and gallotannins” has been substituted by this one: “In vitro, CSAT+® exerts antioxidant capacity that could be due to the presence of several antioxidant compounds such as gallic, gallotannins or phenolic compounds”.
Line 552: As there was not a CSAT+ alone group, you do not know its true effect on weight, only that it did not worsen or improved adiposity and weight gain observed in the MetS model.
The reviewer is right. In order to clarify that this effect is in mice fed a HFHS diet we have modified the sentence by this one: “In vivo, our results show that supplementation with CSAT+® does not affect body weight gain or adiposity in mice fed a HFHS diet.

Reviewer 2 Report
The authors clearly demonstrate the beneficial effects of a carob fruit extract (CSAT+®) on the cardiometabolic alterations associated with metabolic syndrome using both molecular and cellular methods and an animal model. The results in this study strongly support the hypothesis.
The authors can improve the manuscript by modifying two parts:
- On line 153, please describe where the blood was taken. Is it taken from arteries, veins or the decapitated trunk?
- For the TUNEL assay in Figure 8, please provide the actual percentages of TUNEL-positive cells vs the total myocardial cell number. Use the percentage for the statistic analysis
Author Response
Reviewer 2:
The authors clearly demonstrate the beneficial effects of a carob fruit extract (CSAT+®) on the cardiometabolic alterations associated with metabolic syndrome using both molecular and cellular methods and an animal model. The results in this study strongly support the hypothesis.
The authors can improve the manuscript by modifying two parts:
On line 153, please describe where the blood was taken. Is it taken from arteries, veins or the decapitated trunk?
Blood was collected from the decapitated trunk. This information is now stated in the Material and Methods section (Line 150).
For the TUNEL assay in Figure 8, please provide the actual percentages of TUNEL-positive cells vs the total myocardial cell number. Use the percentage for the statistic analysis
It has been corrected as suggested.
